# TLCM: Training-Efficient Latent Consistency Model for Image Generation with 2-8 Steps

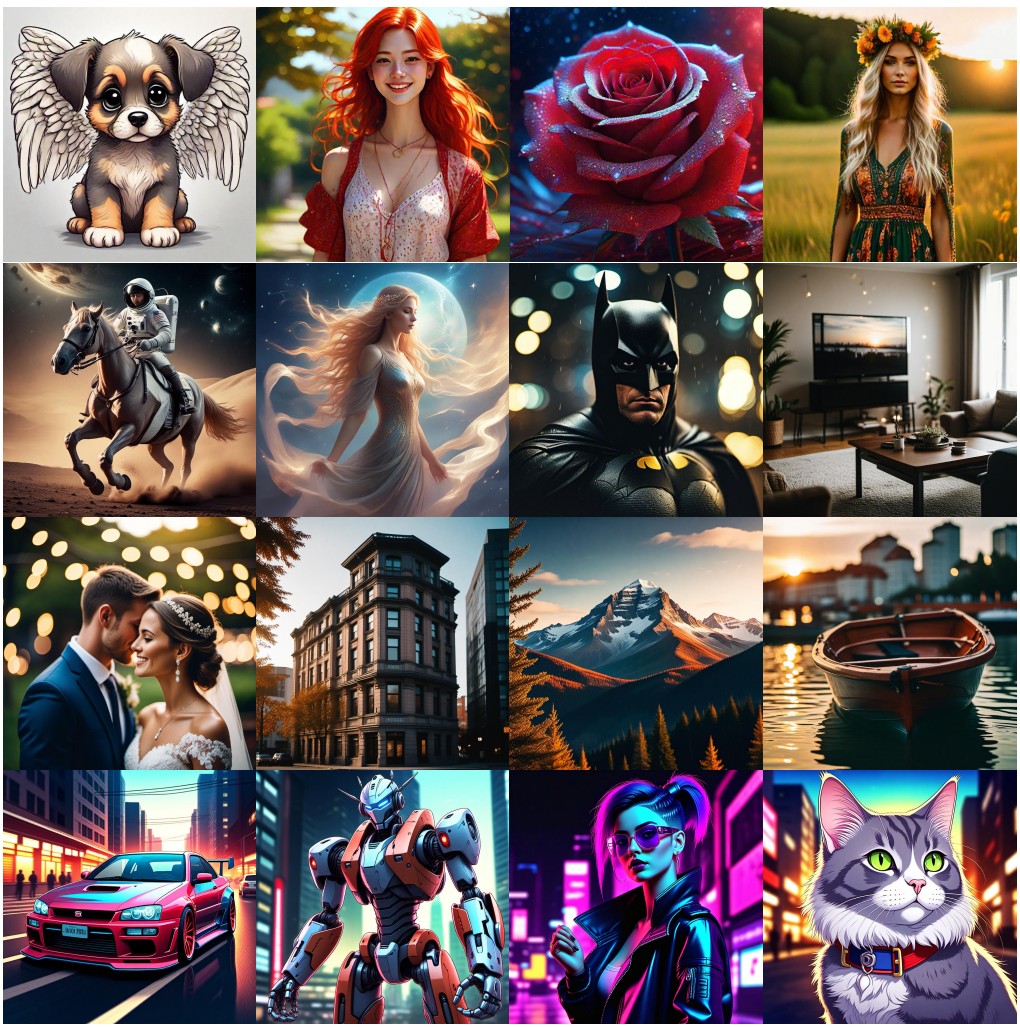

Figure 1: $1024 \times 1024$ image samples from TLCM, distilled from SDXL-base-1.0 (Podell et al.) based on LoRA (Hu et al.). From top to bottom, **2**, **3**, **4** and **8** sampling steps are adopted, respectively. Apart from satisfactory visual quality, TLCM can also yield improved metrics compared to strong baselines.

## Abstract

Distilling latent diffusion models (LDMs) into ones that are fast to sample from is attracting growing research interest. However, the majority of existing methods face two critical challenges: (*i*) They need to perform long-time learning with a huge volume of real data. (*ii*) They routinely lead to quality degradation for generation, especially in text-image alignment. This paper proposes the novel Training-efficient Latent Consistency Model (TLCM) to overcome these challenges. Our method first accelerates LDMs via data-free multistep latent consistency distilla-

tion (MLCD), then data-free latent consistency distillation is proposed to guarantee the inter-segment consistency in MLCD at a low cost. Furthermore, we introduce bags of techniques e.g., distribution matching, adversarial learning, and preference learning, to enhance TLCM's performance at few-step inference without any real data, TLCM demonstrates a high level of flexibility by enabaling adjustment of sampling steps within the range of 2 to 8 while still producing competitive outputs compared to full-step approaches. As the name suggests, TLCM excels in training efficiency in terms of both computational resources and data utilization. Notably, TLCM operates without reliance on a training dataset but instead employs synthetic data for the teacher itself during distillation. With just 70 training hours on an A100 GPU, a 3-step TLCM distilled from SDXL achieves an impressive CLIP Score of 33.68 and an Aesthetic Score of 5.97 on the MSCOCO-2017 5K benchmark, surpassing various accelerated models and even outperforming the teacher model in human preference metrics. We also demonstrate the versatility of TLCMs in applications including image style transfer, controllable generation, and Chinese-to-image generation.

## 1 INTRODUCTION

Diffusion models (DMs) have made great advancements in the field of generative modeling, becoming the go-to approach for image, video, and audio generation (Ho et al., 2020; Kong et al.; Saharia et al., 2022). Latent diffusion models (LDMs) further enhance DMs by operating in the latent image space, pushing the limit of high-resolution image and video synthesis (Ma et al., 2024; Peebles & Xie, 2023; Podell et al.; Rombach et al., 2022). Despite the high-quality and realistic samples, LDMs suffer from frustratingly slow inference–generating a single sample requires tens to hundreds of evaluations of the model, giving rise to a high cost and bad user experience.

There is growing interest in distilling large-scale LDMs into more efficient ones. Concretely, progressive distillation (Lin et al., 2024; Meng et al., 2023; Salimans & Ho, 2023) reduces the sampling steps by half in each turn but finally hinges on a set of models for various sampling steps. InstaFlow Liu et al., UFO-Gen Xu et al. (2024b), DMD (Yin et al., 2024b), and ADD Sauer et al. (2023) target one-step generation, yet losing or weakening the ability to benefit from more (e.g., $> 4$) sampling steps. Latent consistency models (LCMs) Luo et al. (2023) apply consistency distillation (Song et al., 2023) on LDMs' reverse-time ordinary differential equation (ODE) trajectories to conjoin one- and multi-step generation, but the image quality degrades substantially, especially in 2-4 steps. HyperSD (Ren et al., 2024) applies consistency trajectory distillation (Kim et al., 2023) in segments of the ODE trajectory, yet suffers from a substantial performance drop in text-image alignment. Besides, all these methods rely on a huge volume of high-quality data and long training time, hindering their applicability to downstream scenarios with rare compute and data.

Before presenting our proposal, it's essential to note that one-step generation may not always be the optimal choice in practical applications—empirically, sampling with 2-4 steps introduces less than 40% additional computational time compared to one step but can notably enhance the upper limit of sampling quality. Moreover, some practical applications typically have a low tolerance for quality degradation and hence can accept a moderate number of sampling steps. Thereby, this paper aims to develop a unified model with 2-8 sampling steps that can deliver competitive quality comparable to full-step counterparts. We propose **T**raining-efficient **L**atent **C**onsistency **M**odels (TLCMs) to achieve this at the expense of minimal computation and training data. Technically, we introduce data-free multistep latent consistency distillation (MLCD) for fast training at a low cost. After MLCD, we propose a data-free latent consistency distillation (LCD) term for global consistency. To enhance LCD, we enforce the consistency of TLCM at sparse predefined timesteps instead of the entire timestep range, which reduces the learning difficulty of LCD and accelerate convergence. A multistep solver is further explored to unleash the potential of teacher in LCD. Besides, we train a latent LPIPS model to constrain the perceptual consistency of the distilled model in latent space. To optimize TLCM's performance at few-step inference, we explore preference learning, distribution matching, and adversarial learning techniques for regularization in data-free manner.

We have performed comprehensive empirical studies to evaluate TLCMs. We first assess the image quality on the MSCOCO-2017 5K benchmark. We observe the TLCM distilled from SDXL (Podell

et al.) gets an Aesthetic Score (AS) (Schuhmann) of 5.97, and a CLIP Score (CS) (Hessel et al., 2021) of 33.68 with only 3 steps, substantially surpassing 4-step LCM, 8-step SDXL-Lightning (Lin et al., 2024), and 8-step HyperSD, comparable to 25-step DDIM. Additionally, TLCM is obtained by only 70 A100 training hours without any real data, significantly reducing training costs. We also demonstrate the versatility of TLCMs in applications including image stylization, controllable generation, and Chinese-to-image generation.

We summarize our contributions as follows:

- We propose TLCMs to accelerate LDMs to generate high-quality outputs within $2-8$ steps, at the expense of minimal training compute and data.

- We establish a data-free multistep latent consistency distillation and improved latent consistency distillation pipeline for fast LDM acceleration. Besides, bags of data-free techniques are incorporated to boost rare-step quality.

- TLCM achieves a state-of-the-art CS (33.68) and AS (5.97) in 3 steps, surpassing competing baselines, such as 4-step LCM, 8-step SDXL-Lightning, and 8-step HyperSD.

- TLCMs' versatility extends to scenarios such as image stylization, controllable generation, and Chinese-to-image generation, paving the path for extensive practical applications.

## 2 RELATED WORKS

**Diffusion models.** (DMs) (Ho et al., 2020; Sohl-Dickstein et al., 2015; Song & Ermon, 2019; 2020; Song et al., b) progressively add Gaussian noise to perturb the data, then are trained to denoise noise-corrupted data. In the inference stage, DMs sample from a Gaussian distribution and perform sequential denoising steps to reconstruct the data. As a type of generative model, they have demonstrated impressive capabilities in generating realistic and high-quality outputs in text-to-image generation (Podell et al.; Rombach et al., 2022; Saharia et al., 2022), video generation (Peebles & Xie, 2023). To enhance the condition awareness in conditional DMs, the classifier-free guidance (CFG) (Ho & Salimans, 2021) technique is proposed to trade off diversity and fidelity.

**Diffusion acceleration.** The primary challenges that hinder the practical adoption of DMs is the slow inference involving tens to hundreds of evaluations of the model.

Early works like Progressive Distillation (PD) (Salimans & Ho, 2023) and Classifier-aware Distillation (CAD) (Meng et al., 2023) explore the approaches of progressive knowledge distillation to compress sampling steps but lead to blurry samples within four sampling steps. Consistency models (CMs) (Song et al., 2023), consistency trajectory models (CTMs) (Kim et al., 2023) and Diff-Instruct (Luo et al., 2024) distill a pre-trained DM into a single-step generator, but they do not verify the effectiveness on large-scale text-to-image generation.

Recently, the distillation of large-scale text-to-image DMs has gained significant attention. LCM (Luo et al., 2023) extends CM to text-to-image generation with few-step inference but synthesizes blurry images in four steps. InstaFlow (Liu et al.), UFOGen (Xu et al., 2024b), BOOT (Gu et al., 2023), SwiftBrush (Nguyen & Tran, 2024), DMD (Yin et al., 2024a), and Diffusion2GAN Kang et al. (2024) propose one-step sampling for text-to-image generation but are unable to extend their sampler to multiple steps for better image quality.

More recently, SDXL-Turbo (Sauer et al., 2023), SDXL-Lighting (Lin et al., 2024), and HyperSD (Ren et al., 2024) are proposed to further enhance the image quality with few-step sampling but their training depends on huge high-quality text-image pairs and expensive online training. Our method not only enables the generation of high-quality samples using a 2 8 steps sampler but also enhances model performance with more inference cost. Furthermore, our training strategy is resource-efficient and does not require any images.

**Human preference for text-to-image model.** ImageReward (IR) (Xu et al., 2024a) and Aesthetic Score (Schuhmann) are proposed to evaluate the human preference of text-to-image model. Multi-dimensional Preference Score (MPS) (Zhang et al., 2024) improves metrics by learning diverse preferences. To optimize TLCM towards human preference, we incorporate effective reward learning into TLCM to directly guide model tuning.

## 3 PRELIMINARY

### 3.1 DIFFUSION MODELS

Diffusion models (DMs) (Ho et al., 2020; Sohl-Dickstein et al., 2015; Song et al., b) are specified by a predefined forward process that progressively distorts the clean data $x_0$ into a pure Gaussian noise with a Gaussian transition kernel. It is shown that such a process can be described by the following stochastic differential equation (SDE) (Karras et al., 2022; Song et al., b):

$$dx_t = f(x,t)x_t dt + g(t)dw_t, \tag{1}$$

where $t \in [0, T]$, $w_t$ is the standard Brownian motion, and $f(x,t)$ and $g(t)$ are the drift and diffusion coefficients respectively. Let $p_t(x_t)$ denote the corresponding marginal distribution of $x_t$.

It has been proven that this forward SDE possesses the identical marginal distribution as the following probability flow (PF) ordinary differential equation (ODE) (Song et al., b):

$$dx_t = \left[ f(x,t)x_t - \frac{1}{2}g^2(t)\nabla x_t \log p_t(x_t) \right] dt. \tag{2}$$

As long as we can learn a neural model $\epsilon_\theta(x_t, t)$ for estimating the ground-truth score $\nabla x_t \log p_t(x_t)$, we can then draw samples that roughly follow the same distribution as the clean data by solving the diffusion ODE. In practice, the learning of $\epsilon_\theta(x_t, t)$ usually boils down to score matching (Song et al., b).

The ODE formulation is appreciated due to its potential for accelerating sampling and has sparked a range of works on specialized solvers for diffusion ODE (Lu et al., 2022a;b; Song et al., a). Let $\Psi$ denote an ODE solver, e.g., the deterministic diffusion implicit model (DDIM) solver (Song et al., a). The sampling iterates by:

$$x_{t_{n-1}} = \Psi(\epsilon_\theta(x_{t_n}, t_n), t_n, t_{n-1}), \tag{3}$$

where $\{t_n\}_{n=0}^N$ denotes a set of pre-defined discretization timesteps and $t_N = T, t_0 = 0$.

### 3.2 CONSISTENCY MODELS

Consistency model (CM) (Song & Dhariwal; Song et al., 2023) aims at generating images from Gaussian noise within one sampling step. Its core idea is to learn a model $f_\theta(x_t, t)$ that directly maps any point $x_t$ on the trajectory of the diffusion ODE to its endpoint. To achieve this, CMs first parameterizes $f_\theta(x_t, t)$ as:

$$f_\theta(x_t, t) = c_{skip}(t)x_t + c_{out}(t)F_\theta(x_t, t), \tag{4}$$

where $c_{skip}(t), c_{out}(t)$ is pre-defined to guarantee the boundary condition $f_\theta(x_0, 0) = x_0$, and $F_\theta(x_t, t)$ is the neural network (NN) to train.

CM can be learned from a pre-trained DM $\epsilon_{\theta_0}$ via consistency distillation (CD) by minimizing (Song et al., 2023):

$$\mathcal{L}_{CD} = d\big( f_\theta(x_{t_m}, t_m), f_{\theta^-}(x_{t_n}, t_n) \big), \tag{5}$$

where $t_m \sim \mathcal{U}[0, T]$, $x_{t_m} \sim p_{t_m}(x_{t_m})$, $t_n \sim \mathcal{U}[0, t_m]$, $x_{t_n} = \Psi(\epsilon_{\theta_0}(x_{t_m}, t_m), t_m, t_n)$, $d(.,.)$ is some distance function, and $\theta^-$ is the exponential moving average (EMA) of $\theta$. Typically, $x_{t_n}$ is obtained by single-step solver (SS) $\Psi$.

Multistep consistency models (MCMs) (Heek et al., 2024) generalize CMs by splitting the entire range $[0, T]$ into multiple segments and performing consistency distillation individually within each segment. Formally, MCMs first define a set of milestones $\{t_{\text{step}}^s\}_{s=0}^M$ ($M$ denotes the number of segments), and minimize the following multistep consistency distillation (MCD) loss:

$$\mathcal{L}_{MCD} = d\big( \text{DDIM}(x_{t_m}, f_\theta(x_{t_m}, t_m), t_m, t_{\text{step}}^s), \text{DDIM}(x_{t_n}, f_{\theta^-}(x_{t_n}, t_n), t_n, t_{\text{step}}^s) \big), \tag{6}$$

where $s$ is uniformly sampled from $\{0, \ldots, M\}$, $t_m \sim \mathcal{U}[t_{\text{step}}^s, t_{\text{step}}^{s+1}]$, $t_n = t_m - 1$, and $\text{DDIM}(x_{t_m}, f_\theta(x_{t_m}, t_m), t_m, t_{\text{step}}^s)$ means one-step DDIM transformation from state $x_{t_m}$ at timestep $t_m$ to timestep $t_{\text{step}}^s$ based on the estimated clean image $f_\theta(x_{t_m}, t_m)$ (Song et al., a).

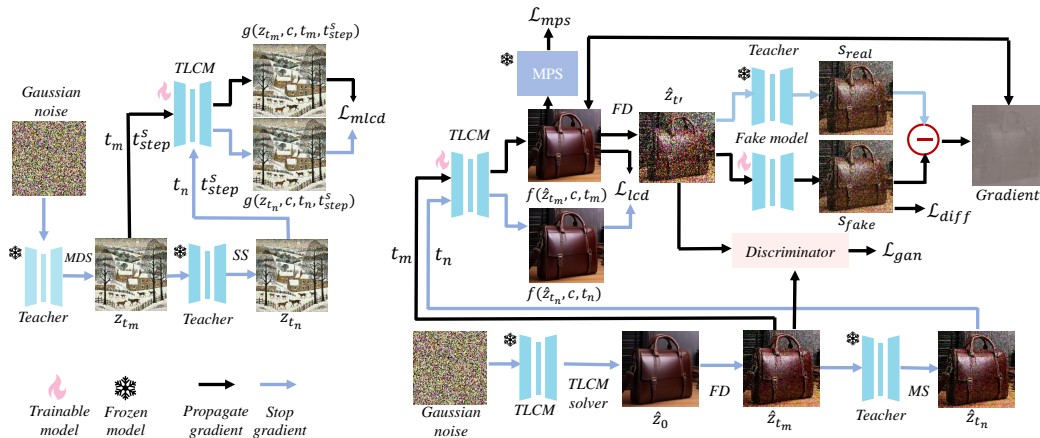

Figure 2: The overview for training TLCM. Data-free multistep latent consistency distillation is first used to accelerate LDM, obtaining initial TLCM (The left part of the overview). Then, data-free latent consistency distillation is proposed to enforce global consistency of TLCM. MPS optimization, DM, and adversarial learning are exploited to promote TLCM's performance in data-free manner (The right part of the overview). Note that we omit Latent LPIPS model for brevity.

## 4 METHODOLOGY

Our target is to accelerate LDM into few-step model, with performance competitive to long-iteration teacher. The learning procedure should be executed with cheap cost in data-free manner. In this section, we propose a novel and unified Training-efficient Latent Consistency Model (TLCM) with 2-8 step's inference. We begin by introducing data-free multistep latent consistency distillation. Subsequently, we discuss data-free latent consistency distillation to enforce global consistency of TLCM. Lastly, we explore various techniques to promote TLCM's performance in data-free manner. The overview of our training pipeline is presented in Figure 2.

### 4.1 DATA-FREE MULTISTEP LATENT CONSISTENCY DISTILLATION

We consider distilling representative pre-trained LDMs, e.g., SDXL (Podell et al.). Previous LCM Luo et al. (2023) has distilled SDXL into few-step model, but it results in the big performance drop, since it is hard to learn the mapping between an arbitrary state of the entire ODE trajectory to the endpoint. Drawing inspiration from MCM, we split the entire range $[0, T]$ into $M$ segments, and then only enforce consistency at each separate segment. To speed up convergence, we change the skipping step $(skip)$ =1 in MCM into 20. The EMA module is removed to save memory consumption. Let $z_t$ denote the states at timestep $t$ in the latent space. We abuse $\epsilon_{\theta_0}(z_t, c, t)$ and $f_\theta(z_t, c, t)$ to denote the pre-trained LDM and target model respectively, where $c$ refers to the generation condition. We formulate the multistep latent consistency distillation (MLCD) loss as:

$$\mathcal{L}_{mlcd} = \|g_\theta(z_{t_m}, t_m, t^s_{step}, c) - \text{nograd}(g_\theta(z_{t_n}, t_n, t^s_{step}, c)\|^2_2, \tag{7}$$

where $g_\theta(z_{t_m}, t_m, t^s_{step}, c) = \text{DDIM}\big(z_{t_m}, f_\theta(z_{t_m}, c, t_m), t_m, t^s_{step}\big)$ represents initial TLCM. Given CFG (Ho & Salimans, 2021) is critical for high-quality text-to-image generation, we integrate it to MLCD by:

$$z_{t_n} = \Psi(\hat{\epsilon}_{\theta_0}(z_{t_m}, c, w, t_m), t_m, t_n), \tag{8}$$

where $\hat{\epsilon}_{\theta_0}(z_t, c, w, t) := \epsilon_{\theta_0}(z_t, \emptyset, t) + w(\epsilon_{\theta_0}(z_t, c, t) - \epsilon_{\theta_0}(z_t, \emptyset, t))$ with $w$ as the guidance scale.

However, this training procedure depends on huge high-quality data, which limits its applicability in the scenarios where such data is inaccessible. To deal with this problem, we propose to draw samples from the teacher model as training data. Specifically, instead of obtaining $z_{t_m}$ via adding noise to $z_0$ as in MCM and HyperSD, we initialize $z_T$ as pure Gaussian noise $\epsilon$ and perform denoising with off-the-shelf ODE solvers based on the teacher model $\epsilon_{\theta_0}$ to obtain $z_{t_m}$. Intuitively, with this strategy, we leverage and distill only the denoising ODE trajectory of the teacher without concerning the forward

one. The latent state $z_{t_m}$ can be acquired from $\epsilon$ by a single denoising step, but we empirically observe that this naive strategy is unable to accelerate LDM with desirable performance, due to poor quality of $z_{t_m}$. Theoretically, $z_{t_m}$ contains less noise for smaller $t_m$. Therefore, we design a multistep denoising strategy (MDS) to predict $z_{t_m}$, which executes more sampling iterations for smaller $t_m$ to get cleaner $z_{t_m}$. At this stage, DDIM solver is used to estimate ODE trajectory and generate samples from pure Gaussian noise. We present the details in Algorithm 1 in Appendix A.3.

## 4.2 Improved Data-free Latent Consistency Distillation

After a round of distillation on $M$ segments, TLCM can naturally produce high-quality samples through $M$-step sampling. However, it is empirically observed that the performance decreases when using fewer steps, which is probably because of the larger discretization error caused by long sampling step sizes. To alleviate this, we advocate explicitly teaching TLCM to capture the mapping between the states that cross segments. Upon this goal, we propose data-free latent consistency distillation to promote the model to be consistent across the predefined timesteps.

We do not compile TLCM to keep consistency across the entire timestep range $[0, T]$ since it is hard to learn the mapping that transforms any point along trajectory into real data. Instead, we improve raw LCD through only keeping consistency at the predefined $M$ timesteps, which makes LCD much easier to learn the mapping. Naturally, the skipping step $skip$ is changed to $T/M$. The big $skip$ offers an additional advantage which further accelerates model convergence. Benefiting from the pre-trained TLCM, we can fast yield clean data $\hat{z}_0$ via few-step ($q$-step) sampler, such as 4 steps, eliminating the requirement of real data. The latent state $\hat{z}_{t_m}$ is obtained by adding noise to $\hat{z}_0$ in the forward diffusion process, where $t_m \in \{t_{\text{step}}^s\}_{s=1}^M$. We formulate this procedure as

$$\hat{z}_{t_m} = FD(TLCM(\epsilon, T, c), t_m), \quad \epsilon \in \mathcal{N}(0, I), \tag{9}$$

where $FD$ and $TLCM$ denote forward diffusion and multistep iterations by TLCM. Then, an ODE solver is used to estimate latent state $\hat{z}_{t_n}$ from $\hat{z}_{t_m}$. Raw LCD adopts one-step solver to predict $\hat{z}_{t_n}$. We argue that it restricts the capability of the teacher due to discretization error, especially for big $skip$. As a result, we explore multistep solver (MS) to unleash the potential of the teacher. Concretely, the time interval $T/M$ is uniformly divided into $p$ parts, and then $p$-step DDIM with CFG is used to calculate $\hat{z}_{t_n}$. The improved data-free LCD loss in stage 2 is:

$$\mathcal{L}_{ilcd2} = \|\big(f_\theta(\hat{z}_{t_m}, c, t_m)\big) - \text{nograd}\big(f_\theta(\hat{z}_{t_n}, c, t_n)\big)\|_2^2. \tag{10}$$

We present the details in Algorithm 2 in appendix A.3. Surprisingly, our improved data-free LCD only costs 2K-iteration training to achieve convergence.

## 4.3 Incorporating Bag of Techniques into TLCM in Data-free Manner

**Latent LPIPS.** Typical LCD directly adopts mean square error loss ($\mathcal{L}_{mse}$) to enforce consistency in the latent space, but it can not capture perceptual features. LPIPS (Zhang et al., 2018) can extract the features matching human perceptual responses. Meanwhile, it has been widely used as an effective regression loss across many image translation tasks. Thereby, we aim to integrate LPIPS into our distillation pipeline to enhance TLCM's performance. However, LPIPS is built in the pixel space, and hence we have to reconstruct latent codes to pixel space to use LPIPS. To reduce training time, we train a latent LPIPS (L-LPIPS) model, which computes perceptual features in latent space. Latent LPIPS model adopts VGG network by changing the input to 4 channels and removing the 3 max-pooling layers, as the latent space in LDM is already 8× downsampled. The model is trained from scratch on BAPPS dataset (Zhang et al., 2018). Base on L-LPIPS, the outputs of the model $g_\theta$ and $f_\theta$ are first fed into L-LPIPS model, whose outputs are used to calculate consistency loss via Equation (7) or Equation (10).

**MPS optimization.** Since TLCMs transform the points on the trajectory to clean samples $\hat{x}_0$, we can naturally directly maximize the feedback of the scorer on the sample $s(\hat{x}_0, c)$. Considering multi-dimensional preference score (Zhang et al., 2024) can measure diverse human preferences, we leverage it to improve TLCM towards human preference. Formally, we optimize the following MPS loss ($\mathcal{L}_{mps}$):

$$\mathcal{L}_{mps} = \max(s_0 - s(\hat{x}_0, c_{pos}), 0) + \max(s(\hat{x}_0, c_{neg}), 0), \tag{11}$$

where $c_{pos}$ represents the text condition corresponding to the images while $c_{pos}$ denotes the irrelevant texts. $\mathcal{L}_{mps}$ maximizes $s(\hat{x}_0, c_{pos})$ with margin $s_0$ and simultaneously minimizes $s(\hat{x}_0, c_{neg})$ with margin 0. The gradients are directly back-propagated from the scorer to model parameters $\theta$ for optimization. We do not use ImageReward or AS to optimize TLCM, because we find IR tends to cause overexposure and AS results in oversaturation for generated images.

**Distribution matching.** Distribution matching (Yin et al., 2024a) is proposed to transform LDM to one-step model. We effectively integrate it into our distillation method to enhance the performance of TLCM. To remove the need of real data, we exploit Equation 9 to get noisy latent $\hat{z}_t$. Data-free DM loss in $\mathcal{L}_{dfdm}$ is applied to optimize TLCM at sparse-step inference as

$$\mathcal{L}_{dfdm} = -\mathbb{E}_{t,\epsilon,\hat{z}_t}[s_{real}(FD(f_\theta(\hat{z}_t, t, c), t')) - s_{fake}(FD(f_\theta(\hat{z}_t, t, c), t'))\nabla_\theta f_\theta(\epsilon)], \qquad (12)$$

where $s_{real}$ and $s_{fake}$ denote the pre-trained score model and fake score model, both initialized by SDXL. The model $s_{fake}$ is finetuned on synthetic data $\hat{z}_0$ through noise prediction loss $\mathcal{L}_{diff}$ in DM (Yin et al., 2024a).

**Adversarial learning.** For high-resolution text-to-image generation, considering the high data dimensionality and complex data distribution, simply using MSE loss fails to capture data discrepancy precisely, thus providing imperfect consistency constraints. We propose to use GAN loss to enforce the distribution consistency. Unlike previous methods needing real data to execute adversarial learning, we exploit Equation 9 to obtain $\hat{z}_t$. The student model $f_\theta$ denoises $\hat{z}_t$ by one step, obtaining $\widetilde{z}_0$. Through discriminator $D$, the GAN loss $\mathcal{L}_{gan}$ is formulated as

$$\mathcal{L}_{gan} = \log(D(FD(\hat{z}_0, t'))) - \log(D(FD(\widetilde{z}_0, t'))). \qquad (13)$$

## 5 EXPERIMENTS

### 5.1 MAIN RESULTS

We quantitatively compare our method with both the DDIM (Song et al., a) baseline and acceleration approaches including LCM (Luo et al., 2023), SDXL-Turbo (Sauer et al., 2023), SDXL-Lightning (Lin et al., 2024), HyperSD (Ren et al., 2024), CS (Hessel et al., 2021) with ViT-g/14 backbone, AS (Schuhmann), IR (Xu et al., 2024a), Fréchet Inception Distance (FID) are exploited as objective metrics. The evaluation is performed on MSCOCO-2017 5K validation dataset (Lin et al., 2014). All methods perform zero-shot validation except for HyperSD since it utilizes the MSCOCO-2017 dataset for training. Only SDXL-Turbo produces 512-pixel images while the others generate 1024-pixel images. We only report FID for reference and do not analyze it since FID on COCO is not reliable to evaluate text-to-image models (Sauer et al., 2023; Ren et al., 2024).

The metrics of various methods are listed in Table 1. We use "-" to represent metric when it is missing in the corresponding paper. We can observe that our TLCM only costs 70 A 100 training hours, even without any data. Compared to other methods, TLCM significantly reduces training resources, which is very valuable for most laboratories and the scenarios when real data are inaccessible. our 3-step TLCM presents superior CS, AS, IR than 4-8 step's LCM (Luo et al., 2023), SDXL-Lightning (Lin et al., 2024). These results indicate our TLCM's synthetic images are much better aligned with texts and the human preference than LCM, SDXL-Lightning. Excitingly, our 3-step TLCM outperforms 25-step teacher in terms of AS and IR, and achieves comparable CS value, demonstrating TLCM almost reserves all the information in teacher and even introduces new human preference knowledge via the proposed distillation method. Our 3-step TLCM shows much higher CS than 4-8 step's HyperSD, indicating HyperSD loses much information in the distillation procedure, because it fails to sufficiently ensure consistency constraint. We notice IR value of HyperSD is higher than our TLCM. This is because IR model has been used to optimize HyperSD. Moreover, we can see the performance of SDXL-Turbo drop with respect to CS and IR when increasing sampling steps. This is because it is designed for specific steps. Instead, our TLCM can improve at least one metric with additional steps. This is valuable since image quality is the primary consideration when affordable computation resource is determined in real applications.

We present the visual comparisons in Figure 3. Under the same conditions, we observe that the images generated by TLCM have better image quality and maintain higher semantic consistency on more challenging prompts, which also leads to greater human preference.

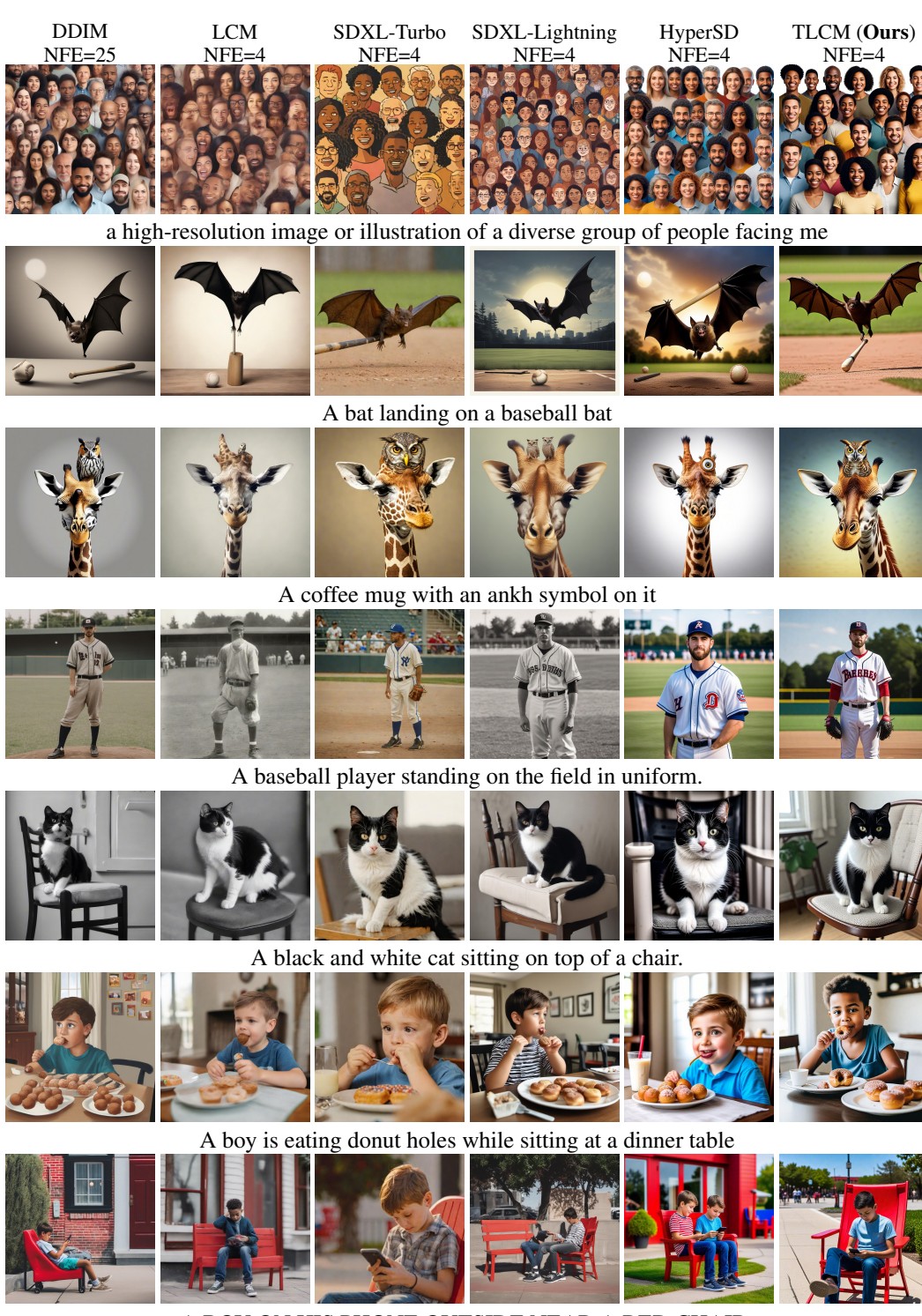

Figure 3: Visual comparison between our TLCM and the state-of-the-art methods. Zoom in for more details.

Table 1: Zero-shot performance comparison on MSCOCO-2017 5K validation datasets with the state-of-the-art methods. All models adopts SDXL architecture. Time: inference time (second) on A100. TH: Training hours using A100. TI: Training images.

| Method | Step | Time | FID | CS | AS | IR | TH | TI |
|---|---|---|---|---|---|---|---|---|
| DDIM | 25 | 3.29 | 23.29 | 33.97 | 5.87 | 0.82 | 0 | 0 |
| LCM | 4 | 0.71 | 27.09 | 32.53 | 5.85 | 0.51 | - | - |
| SDXL-Turbo | 4 | 0.38 | 28.52 | 33.35 | 5.64 | 0.83 | - | - |
| SDXL-Turbo | 8 | 0.61 | 29.64 | 32.81 | 5.78 | 0.82 | - | - |
| SDXL-Lightning | 4 | 0.71 | 27.90 | 32.90 | 5.63 | 0.72 | - | >12M |
| SDXL-Lightning | 8 | 0.99 | 27.04 | 32.74 | 5.95 | 0.71 | - | >12M |
| HyperSD | 4 | 0.71 | 34.45 | 32.64 | 5.52 | 1.15 | 600 | >12M |
| HyperSD | 8 | 0.99 | 35.94 | 32.41 | 5.83 | 1.14 | 600 | >12M |
| TLCM | 2 | 0.58 | 27.50 | 33.18 | 5.90 | 0.97 | 70 | 0 |
| TLCM | 3 | 0.65 | 29.12 | 33.68 | 5.97 | 1.00 | 70 | 0 |
| TLCM | 4 | 0.71 | 30.33 | 33.52 | 6.06 | 1.01 | 70 | 0 |
| TLCM | 5 | 0.78 | 30.90 | 33.69 | 6.04 | 1.01 | 70 | 0 |
| TLCM | 6 | 0.85 | 30.98 | 33.71 | 6.07 | 1.01 | 70 | 0 |
| TLCM | 8 | 0.99 | 32.40 | 33.53 | 6.08 | 1.02 | 70 | 0 |

Table 2: Ablation study of TLCM with respect to latent LPIPS, data-free LCD with single denoising step ($\mathcal{L}_{lcd-s}$), data-free MLCD with single denoising iteration ($\mathcal{L}_{mlcd-s}$), data-free MLCD with MDS ($\mathcal{L}_{mlcd-m}$), data-free LCD in stage 2 ($\mathcal{L}_{lcd2}$), improved data-free LCD in stage 2 ($\mathcal{L}_{ilcd2}$), data-free DM ($\mathcal{L}_{dfdm}$), multi-dimensional human preference ($\mathcal{L}_{mhp}$), adversarial ($\mathcal{L}_{gan}$). All the models adopt 4-step sampler and SDXL backbone.

| L-LPIPS | $\mathcal{L}_{lcd-s}$ | $\mathcal{L}_{mlcd-s}$ | $\mathcal{L}_{mlcd-m}$ | $\mathcal{L}_{lcd2}$ | $\mathcal{L}_{ilcd2}$ | $\mathcal{L}_{mhp}$ | $\mathcal{L}_{dfdm}$ | $\mathcal{L}_{gan}$ | CS | FID | AS | IR |
|---|---|---|---|---|---|---|---|---|---|---|---|---|
| | ✓ | | | | | | | | 31.61 | 32.90 | 5.89 | 0.41 |
| | | ✓ | | | | | | | 31.76 | 27.01 | 5.98 | 0.58 |
| ✓ | | ✓ | | | | | | | 31.99 | 27.61 | 5.92 | 0.61 |
| ✓ | | | ✓ | | | | | | 32.31 | 30.99 | 6.01 | 0.69 |
| ✓ | | | ✓ | ✓ | | | | | 32.74 | 32.05 | 6.00 | 0.72 |
| ✓ | | | ✓ | | ✓ | | | | 33.06 | 25.44 | 5.96 | 0.77 |
| ✓ | | | ✓ | | ✓ | ✓ | | | 33.16 | 28.40 | 6.01 | 0.90 |
| ✓ | | | ✓ | | ✓ | ✓ | ✓ | | 33.32 | 30.58 | 6.03 | 0.97 |
| ✓ | | | ✓ | | ✓ | ✓ | ✓ | ✓ | 33.52 | 30.33 | 6.06 | 1.01 |

## 5.2 ABLATION STUDY

To analyze the key components of our method, we make a thorough ablation study to verify the effectiveness of the proposed TLCM. Table 2 depicts the performance of TLCM's variants.

**Data-free multistep latent consistency distillation.** As shown in Table 2, only using $\mathcal{L}_{lcd-s}$ which computes $z_{t_m}$ by single step for LCD achieves CS score of 31.61, AS of 5.89, indicating our data-free method is able to accelerate LDM with good quality. Changing $\mathcal{L}_{lcd-s}$ to single-step denoising MLCD $\mathcal{L}_{mlcd-s}$, all metrics are improved. This result verifies that MLCD has a stronger capability to accelerate LDM than LCD. This is because it is hard for data-free LCD to enforce consistency across the entire timestep range while data-free MLCD alleviates this by performing LCD within predefined multiple segments.

**Denoising strategy.** We can observe from Table 2 that $\mathcal{L}_{mlcd-m}$ substantially enhances the performance of $\mathcal{L}_{mlcd-s}$ , verifying that the proposed multistep denoising strategy is critical to perform data-free MLCD. The probable reason is our multistep MDS yield better initial latent codes, where the latent codes have better quality with smaller timesteps.

Table 3: Performance comparison of the teacher's sampling steps for data-free consistency distillation in stage 2.

| Step | CS | FID | AS | IR | Step | CS | FID | AS | IR |
|---|---|---|---|---|---|---|---|---|---|
| 1 | 32.78 | 26.19 | 5.95 | 0.66 | 2 | 32.97 | 25.73 | 5.95 | 0.71 |
| 3 | 33.06 | 25.44 | 5.96 | 0.77 | 4 | 33.10 | 25.18 | 5.97 | 0.78 |

**Latent LPIPS.** As outlined in Table 2, $\mathcal{L}_{mlcd-s}$ using L-LPIPS introduces gains on all metrics. This result denotes it is more powerful to enforce consistency in latent LPIPS space than raw latent space as latent LPIPS can make perceptual consistency.

**Data-free latent consistency distillation in stage 2.** In 2, $\mathcal{L}_{lcd2}$ represents using mulltistep solver in LCD to enforce consistency across the entire timestep range. We can see $\mathcal{L}_{lcd2}$ significantly improves CS values of TLCM trained in stage 1. This is because $\mathcal{L}_{lcd2}$ achieves inter-segment consistency of TLCM. The performance is further enhanced by substituting $\mathcal{L}_{lcd2}$ with $\mathcal{L}_{ilcd2}$. The reason lies in that it is easier to make consistency along the sparse predefined timesteps than the entire timestep range.

**MHP optimization.** Table 2 shows that adding $\mathcal{L}_{mhp}$ to the losses in line 7 introduces gains in terms of CS and IR. This result indicates that our MHP optimization method is capable of improving the text-image alignment and human preference of TLCM.

**Data-free DM.** We can see in Table 2 using our data-free DM loss $\mathcal{L}_{dfdm}$ leads to the performance improvements on all metrics. This result demonstrates that our DM in data-free way is compatible to the proposed distillation method, boosting TLCM's performance.

**Discriminator.** We also observe in Table 2 that discriminator loss $\mathcal{L}_{gan}$ improves CS, AS, and IR, because discriminator facilitate consistency in probability distribution space, which is critical for low-step regime.

**Teacher's inference steps of data-free latent consistency distillation in stage 2.** In Table 3, we study the effect concerning teacher's sampling steps of data-free LCD in stage 2. The results shows as sampling step increases from 1 to 4, the performance is consistently improved. Therefore, it is crucial to perform multi-step denoising to estimate $\hat{z}_{t_n}$. The reason is that multi-step solvers is capable of reducing discretization error for big skipping step.

## 6 CONCLUSION

In this paper, we propose Training-efficient Latent Consistency Model (TLCM), a novel approach for accelerating text-to-image latent diffusion models using only 70 A100 hours, without requiring any text-image paired data. TLCM can generate high-quality, delightful images with only 2-8 sampling steps and achieve better image quality than baseline methods while being compatible with image style transfer, controllable generation, and Chinese-to-image generation.

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

# A APPENDIX

## A.1 IMPLEMENTATION DETAILS

We use the prompts from LAION-Aesthetics- 6+ subset of LAION-5B Schuhmann et al. (2022) to train our model. We train the model with 12000 iterations for data-free MLCD and 2000 iterations for data-free LCD. After LCD, MPS optimization runs 500 iterations with batch size of 8. Then, DM and adversarial learning are used to improve TLCM with 1000 iterations with batch size of 4. The whole procedure uses Adamw optimizer and 4 A100. Only MLCD adopts learning rate of 1e-4 and the other stages use learning rate of 1e-5. The discriminator adopts learning rate of 1e-4 and AdamW optimizer. The initial segment number $M$ is 8 and $s_0$ for MPS is 16. We set the guidance scale $w$ in CFG as 8.0, the denoising steps $p = 3$ for teacher to compute $\hat{z}_{t_n}$, and $q = 4$ for TLCM to compute $\hat{z}_0$. As for model configuration, we use SDXL Podell et al. as teacher to estimate trajectory while student model $f_\theta$ is also initialized by SDXL. The discriminator is also initialized by SDXL. We train a unified Lora instead of UNet in all the distillation stages for convenient transfer to downstream applications.

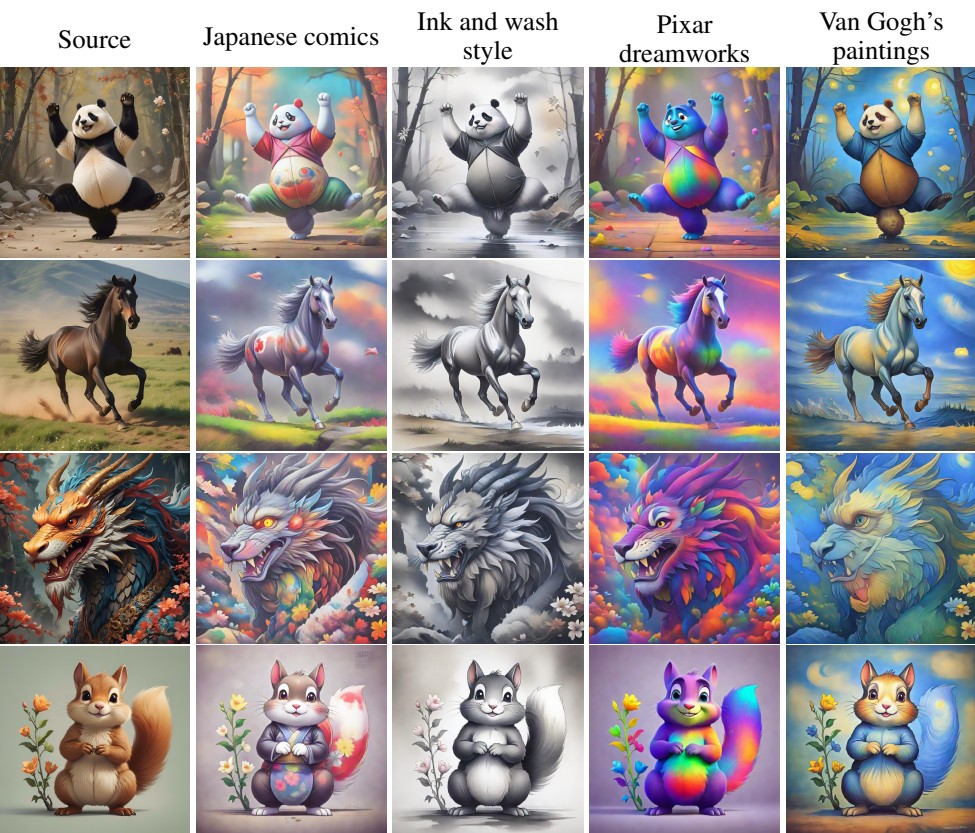

Figure 4: TLCM with image style transfer. The styles are presented at the top, and we apply image style transfer on the source image with our TLCM. Two-step sampling can produce highly stylized images with excellent results.

## A.2 APPLICATION

### A.2.1 ACCELERATION OF IMAGE STYLE TRANSFER

Our TLCM LoRA is compatible with the pipeline of image style transfer (Mou et al., 2024). We present some examples in Figure 4 with only 2-step sampling.

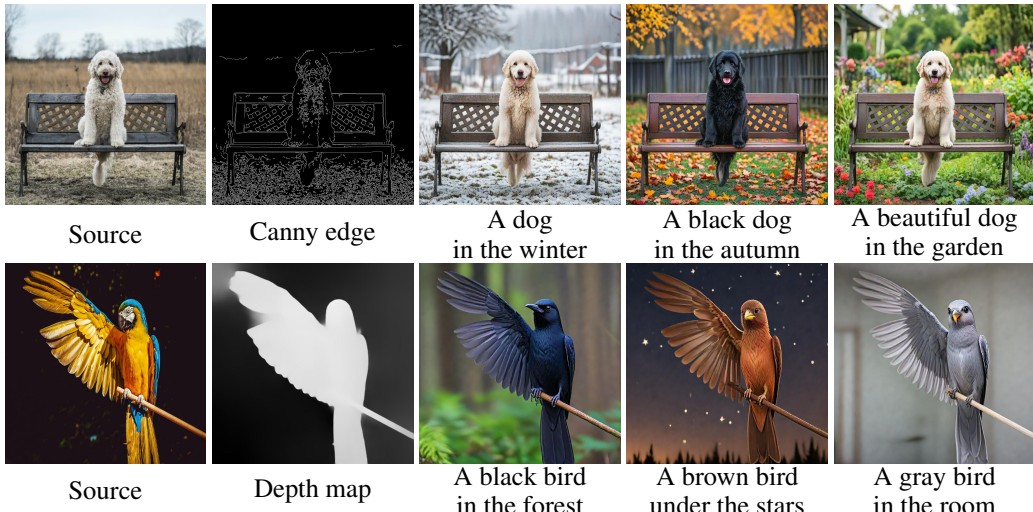

| Source | Canny edge | A dog in the winter | A black dog in the autumn | A beautiful dog in the garden |
| Source | Depth map | A black bird in the forest | A brown bird under the stars | A gray bird in the room |

Figure 5: TLCM with ControlNet. Our TLCM can be incorporated into ControlNet pipeline and produce satisfactory results with 2 steps sampling.

### A.2.2 ACCELERATION OF CONTROLLABLE GENERATION

Our TLCM LoRA is compatible with Controlnet, enabling accelerated controllable generation. We utilize canny and depth ControlNet based on SDXL-base, together with TLCM LoRA in Figure 5. The results are sampled in 2 steps. We observe our model achieves superior image quality and demonstrates compatibility with other models, e.g. ControlNet, while also providing enhanced acceleration capabilities.

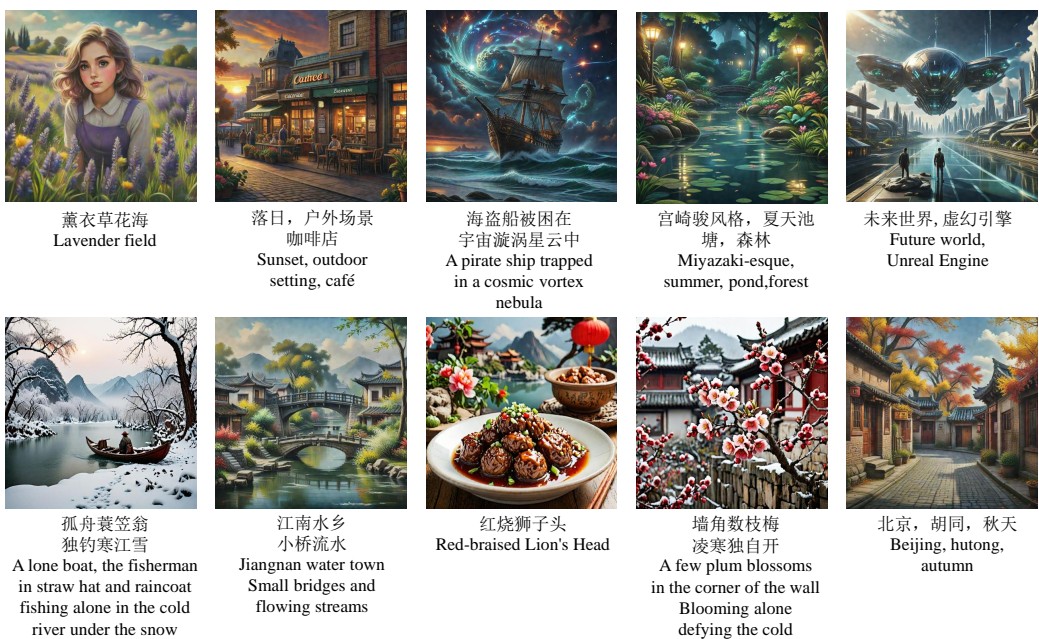

| 薰衣草花海 Lavender field | 落日，户外场景咖啡店 Sunset, outdoor setting, café | 海盗船被困在宇宙漩涡星云中 A pirate ship trapped in a cosmic vortex nebula | 宫崎骏风格，夏天池塘，森林 Miyazaki-esque, summer, pond, forest | 未来世界, 虚幻引擎 Future world, Unreal Engine |
| 孤舟蓑笠翁 独钓寒江雪 A lone boat, the fisherman in straw hat and raincoat fishing alone in the cold river under the snow | 江南水乡 小桥流水 Jiangnan water town Small bridges and flowing streams | 红烧狮子头 Red-braised Lion's Head | 墙角数枝梅 凌寒独自开 A few plum blossoms in the corner of the wall Blooming alone defying the cold | 北京，胡同，秋天 Beijing, hutong, autumn |

Figure 6: TLCM for Chinese-to-image generation. With 3 steps sampling, our TLCM model can produce images that align with Chinese semantic meaning. The first line presents images in general Chinese contexts, while the second line showcases images in specific Chinese cultural settings.

### A.2.3 ACCELERATION OF CHINESE-TO-IMAGE GENERATION

Our TLCM can accelerate the generation speed of Chinese-to-image diffusion model (Ma et al., 2023). We present some examples in Figure 6.

### A.3 ALGORITHMS

---

**Algorithm 1:** Data-free multistep latent consistency distillation

---

**Input:** Gaussian noise $\epsilon$, timestep $t_m$, segment index $s$, teacher model $\epsilon_{\theta_0}$, student model $g_\theta$, text condition $c$, segment number $M$

Initialize $z_T$ with $\epsilon$, calculate denoising steps $L = M - s$, time interval $\triangle T = (T - t_m)/L$

**for** $i$ $in$ $\{0, 1, \cdots, L-1\}$ **do**
  Calculate $t = T - i * \triangle T$, $t_{m'} = t - \triangle T$
  Calculate $z_{t_{m'}} = \Psi(\hat{\epsilon}_{\theta_0}(z_t, c, w, t), t, t_{m'})$
**end**

Calculate $z_{t_n}$ using Equation (8)

Perform MLCD using Equation (7)

---

**Algorithm 2:** Data-free latent consistency distillation in stage 2

---

**Input:** Gaussian noise $\epsilon$, timestep $t_m$, teacher model $\epsilon_{\theta_0}$, student model $f_\theta$, text condition $c$, segment number $M$, denoising step of teacher $p$, denoising step $q$ of student, diffusion coefficient sequence $\alpha_{1:T}$, timestep milestones $\{t_{step}^s\}_{s=0}^M$

Initialize $\hat{z}_T$ with $\epsilon$ and timestep $t$ with $T$

**for** $i$ $in$ $\{0, 1, \cdots, q-1\}$ **do**
  Calculate $\hat{z}_0 = \dfrac{\hat{z}_t - \sqrt{1 - \alpha_t} f_\theta(\hat{z}_t, t, c)}{\sqrt{\alpha_t}}$
  Calculate $t = T - T/q \times (i+1)$, Calculate $\hat{z}_t = \sqrt{\alpha_t}\hat{z}_0 + \sqrt{1 - \alpha_t}\epsilon$
**end**

Randomly sample $t_m$ from $\{t_{step}^s\}_{s=1}^M$, detach $\hat{z}_0$ and calculate $\hat{z}_{t_m}$ by forward diffusion
$\hat{z}_{t_m} = \sqrt{\alpha_{t_m}}\hat{z}_0 + \sqrt{1 - \alpha_{t_m}}\epsilon$

**for** $i$ $in$ $\{0, 1, \cdots, p-1\}$ **do**
  Calculate $t_1 = t_m - (T/M)/p \times i$ and $t_2 = t_m - (T/M)/p \times (i+1)$
  Calculate $\hat{z}_{t_2}$ using Equation (8) based on current state $\hat{z}_{t_1}$
**end**

Perform LCD using Equation (10)

---

