# OpenReview forum: "TLCM: Training- efficient Latent Consistency Model for Image Generation with 2-8 Steps"
_ICLR.cc/2025/Conference — ICLR 2025 Conference Withdrawn Submission_

### Official Review · Reviewer_9DDT · 2024-11-02

**Soundness:** 2
**Presentation:** 3
**Contribution:** 1
**Rating:** 3
**Confidence:** 4

**Summary:**

This paper presents a method for latent diffusion model distillation. The core idea of this paper is mainly inspired by latent consistency distillation. The authors implemented some improvements based on the previous success of diffusion distillation, such as sparse timesteps. To prevent the cost of collecting real-world data, the proposed method is trained on generated data from the teacher model instead. A bag of independent techniques such as adversarial training are further included to help the model match the SoTA performance. The evaluations are performed on SDXL.

**Strengths:**

The paper presents the method in a very intuitive way with very informative equations and figures. The overall writing quality of this paper is good.

**Weaknesses:**

- Novelty

This paper presents insufficient novelties. The entire method seems to be an ensemble of diffusion distillation techniques already proven effective. For example, this paper advocates the advantage of relying on no training data and uses synthesized data instead. However, this has already been proven useful in SD3-turbo [1], and learning from generated data is by no means a new idea under the context of model distillation.

- Empirical evaluations

This paper is presented with insufficient evaluations. SDXL, which is a relatively old model, is the only one evaluated in the paper. And there are some important comparisons missing such as SD3-turbo, DMD, and instaflow.

The authors claim, 'We only report FID for reference and do not analyze it since FID on COCO is not reliable to evaluate text-to-image models.' I agree that FID is not the best way for diffusion evaluation. However, the values do reveal something about the proposed methods. I noticed that the FID values increase with the number of steps for the proposed method. In my understanding, this is a result of learning distillation from generated data instead of real data, as the modeled distribution deviates further from the real-world data distribution.

- Supports to claims

I find some claims in the paper require further support. For example, the authors claimed 'They (other distillation methods) need to perform long-time learning with a huge volume of real data.' I find it very hard to agree with this. According to my experience, methods such as LCM can be trained very fast with a small portion of training data. The authors are strongly encouraged to fill in the 'TH' value for LCM in Table 1 for a direct comparison. For the rest of the metrics, I don't see how the proposed method clearly outperforms the rest, except for IR, which is a score that the model is explicitly optimized for.


[1] Fast High-Resolution Image Synthesis with Latent Adversarial Diffusion Distillation, arxiv.

**Questions:**

1. Does the 70GPU-hour training time include the time spent on data generation? The overall cost of data generation is needed to justify L241 'with cheap cost in data-free manner.'

---

### Official Review · Reviewer_C2ZP · 2024-11-02

**Soundness:** 3
**Presentation:** 3
**Contribution:** 2
**Rating:** 5
**Confidence:** 5

**Summary:**

- The authors propose a multi-step distribution matching diffusion model to generate samples within few steps. By dividing the whole iterations into several predefined steps, the consistency model can focus on matching distribution in narrower range.
- Adversarial loss, latent-LPIPS loss, and human preference loss have also been adopted to boost the performance.

**Strengths:**

- Explanation about existing methods have been given enough. It is easy to follow and the application of each component sounds reasonable.
- Resulting images with only 4 steps look quite impressive.
- Necessary ablation studies to make this method solid have been provided.

**Weaknesses:**

- Even though the results are quite impressive, this method is a combination of Multistep Consistency Models(MCM), Distribution Matching Distillation(DMD), and DiffusionGAN. It is hard to ignore novelty issue and this would be the main reason of my decision.
- Too many notations made me really hard to follow this paper. It would be much better if some expressions can be trimmed.

Minor issue
- The caption of the 3rd row in Fig.3 should be modified.

**Questions:**

- After the emergence of rectified flow transformer, I think these kinds of distillation methods should be re-assessed. Apart from the non-predictable nature of the original diffusion trajectory, we now can learn the straightened flow of latent space and the distillation must be a lot easier. Even in the paper of SD3, it says that if the model capacity is given enough, reducing timesteps into a small number sacrifices little performance. This is not a critical comment about this paper and I would recommend the authors to think about this aspect.

---

### Official Review · Reviewer_T6B3 · 2024-11-03

**Soundness:** 2
**Presentation:** 2
**Contribution:** 2
**Rating:** 5
**Confidence:** 4

**Summary:**

Existing methods for distilling the pretrained knowledge of Diffusion Models have several drawbacks; expensive training time, necessity of large scale real data, and image generation quality given a few steps to generate an image. To this end, the authors propose a data-free distillation pipeline named as Training Efficient Latent Consistency Model (TLCM). Briefly, the initial parameter of TLCM is obtained by the proposed first process, and the second process boosts global consistency of TLCM. The experiment results show that TLCM has benefits over the baselines for the short training time and the data-free mechanism.

**Strengths:**

- Interesting idea to use synthetic data for distillation.
- Reasonable motivations.

**Weaknesses:**

- Overview figure is very complex to understand. My suggestion is that it would be better if the right figure is decomposed into two parts; section 4.2 and section 4.3
- Weak explanations on the experiment settings.
    - What is the meaning of each measure?
    - Is higher better or lower better?
    - How many samples are used?
    - What was the input to generate the samples to measure?
    - It sounds more reasonable if the proposed methods contain only IS or FID like [1]. The aim of the work is to distill the knowledge of the pretrained models and reduce the required sampling steps to some extent. Shouldn’t it enough If the generated images by TLCM  (2-8 steps) are as realistic as DDIM? Please provide justifications why the other metrics are needed in detail.
- Weak interpretations of the experiment results.
    - (L364-L365) The proposed methods use synthetic data to distill the pretrained knowledge while the baselines are using real data. How does the proposed method show better performance than the baselines? It sounds more reasonable if the baseline is a sort of upper bound.
    - FID of TLCM (2 steps) is better than FID of TLCM (8 steps). It also shows the pattern that the # of steps and FID are inversely proportional, which is not reasonable.
    - Prioritize what information is important for each table and emphasize those numbers.

- Too many engineering techniques are applied for improving the marginal performance. They dilute the main point of the paper..

Overall, the proposed methods are interesting, but the results (FID) are not convincing enough.

[1] PROGRESSIVE DISTILLATION FOR FAST SAMPLING OF DIFFUSION MODELS, ICLR’22

**Questions:**

(L144-145) This is a vague statement. What is the meaning of the effectiveness?

---

### Official Review · Reviewer_DFJT · 2024-11-04

**Soundness:** 3
**Presentation:** 2
**Contribution:** 2
**Rating:** 5
**Confidence:** 4

**Summary:**

This paper presents a direct combination of Multistep Consistency Models [1] and Latent Consistency Models [2], resulting in a Multistep Latent Consistency Model.

Advantages:
1. While straightforward, the idea shows some novelty
2. The writing is acceptable, though the innovation isn't immediately apparent

Disadvantages:
1. The experimental results appear counterintuitive: contrary to normal expectations where more sampling steps yield better results, this paper shows deteriorating FID scores with increased sampling steps
2. The paper lacks directness - the core innovations are hard to identify, as the authors seem to attempt highlighting multiple contribution points
3. The generated images predominantly show a cyberpunk style, lacking photorealism and overall quality

Technical Issues:
1. Equation 1 appears incorrect - contains an extra x_t term
   Should be: dx_t = f(x_t, t)dt + g(t)dw_t

2. Results Inconsistencies:
   - Tables 1 and 3 show FID increasing with more sampling steps, contradicting common understanding
   - Ablation study shows puzzling results: combining all techniques significantly worsens FID scores
   - Equations 11-13 look unnecessary, just traditional confrontational losses.

Writing Concerns:
The paper lacks focus on core contributions. It is hard for me to distinguish core contributions.

References:
[1] Heek et al., "Multistep consistency models," 2024
[2] Luo et al., "Latent consistency models," 2023
[3] Song et al., "Consistency models," 2023

**Strengths:**

Advantages:
1. While straightforward, the idea shows some novelty
2. The writing is acceptable, though the innovation isn't immediately apparent

**Weaknesses:**

Disadvantages:
1. The experimental results appear counterintuitive: contrary to normal expectations where more sampling steps yield better results, this paper shows deteriorating FID scores with increased sampling steps
2. The paper lacks directness - the core innovations are hard to identify, as the authors seem to attempt highlighting multiple contribution points
3. The generated images predominantly show a cyberpunk style, lacking photorealism and overall quality

**Questions:**

see above

---

### Official Review · Reviewer_Hh6Q · 2024-11-04

**Soundness:** 2
**Presentation:** 1
**Contribution:** 1
**Rating:** 3
**Confidence:** 4

**Summary:**

The authors propose TLCM (Training-efficient Latent Consistency Model) for accelerating text-to-image latent diffusion models in a data-free manner with a small amount of training time. The key innovation is a two-stage distillation process. A data-free multistep latent consistency distillation (MLCD) is proposed to accelerate the base model, followed by another improved data-free latent consistency distillation to ensure global consistency. The authors enhance the model's performance through several techniques including latent LPIPS for perceptual consistency, multi-dimensional preference learning, distribution matching, and adversarial learning. Their empirical results show that TLCM can generate high-quality images in just 2-8 inference steps.

**Strengths:**

-	The authors propose a two-stage distillation scheme that progressively distills the model to a few-step regime with reduced training costs.
-	The proposed method achieves superior results with SDXL on COCO-5k generation.

**Weaknesses:**

-	The submission is basically a technical report on achieving the SOTA few-step image generation performances with data-free multi-step consistency distillation, which is a successful practical combination of [1] and MCM. The second distillation stage also uses a series of SOTA techniques including ADD, DMD, and preference tuning. There are very limited scientific insights or technical innovations.
-	The current presentation is poor and needs substantial improvement. For instance, there is extensive and unnecessary usage of acronyms, making the submission so confusing and hard to read.
-	Limited empirical evaluation. The method is only validated on SDXL. How does it perform on other diffusion models with different architectures, e.g., Pixart?

References

1.	Kohler, Jonas, et al. "Imagine flash: Accelerating emu diffusion models with backward distillation." arXiv preprint arXiv:2405.05224 (2024).

**Questions:**

As in weaknesses.

---

### Note · Authors · 2024-11-15

I have read and agree with the venue's withdrawal policy on behalf of myself and my co-authors.